# PyDoseRT: A physics-informed, plug-and-play dose engine for gradient-based radiotherapy treatment planning

**Attila Simkó**[1]                                        ATTILA.SIMKO@UMU.SE
**Matthias Kronsteiner**[2]
**Simon Glatzer**[2]
**Minh Vu**[1]
**Josef A. Lundman**[1]
**Joakim Jonsson**[1]
**Jörgen Olofsson**[1]
**Kristina Sandgren**[1]
**Wolfgang Lechner**[2]
**Dietmar Georg**[2]
**Tommy Löfstedt**[3]
**Tufve Nyholm**[1]
**Anders Garpebring**[1]
**Gerd Heilemann**[2]

[1] *Department of Diagnostics and Intervention, Umeå University, Umeå, Sweden*

[2] *Department of Radiation Oncology, Medical University of Vienna, Vienna, Austria*

[3] *Department of Computing Science, Umeå University, Umeå, Sweden*

## Abstract

We present PyDoseRT, a GPU-accelerated, differentiable dose engine implemented in PyTorch that computes 3D dose distributions from machine-deliverable parameters (MLC leaf positions, jaw settings, gantry angles, monitor units) while preserving gradients throughout. The engine was validated on 181 clinical VMAT prostate plans from two institutions, achieving mean gamma pass rates of 99.6% and 97.5% (2%/2 mm). We further trained a deep learning model on the LUND-PROBE dataset to predict delivered dose from patient anatomy, using PyDoseRT as a differentiable layer for end-to-end training. The model inherently produced deliverable plans with Dice coefficients of $0.87 \pm 0.02$ and $0.92 \pm 0.03$ for the 50% and 95% isodose volumes on a held-out validation set. PyDoseRT enables TPS-independent, gradient-based radiotherapy optimization and provides a platform for deep learning-based treatment planning.

**Keywords:** Radiotherapy, dose engine, differentiable physics, treatment planning, deep learning

## 1. Introduction

Radiotherapy treatment planning involves optimizing thousands of parameters—multileaf collimator (MLC) positions, jaw positions, gantry angles, monitor units—to achieve clinically acceptable dose distributions. This optimization currently requires expert manual adjustments in commercial treatment planning systems (TPS). The rising global cancer burden (Abdel-Wahab et al., 2024; Aggarwal et al., 2023) and the adoption of image-guided radiotherapy (Qiu et al., 2023) demand automated workflows to replace manual planning.

Deep learning (DL) has shown promise for dose prediction (Zimmermann et al., 2021; Gao et al., 2025), but predicted dose maps are not deliverable plans and must be imported into a TPS for aperture optimization (Mekki et al., 2025; Chang et al., 2025). Parameter prediction methods bypass the TPS but do not model how parameter changes affect dose (Heilemann et al., 2023, 2025). A differentiable dose engine bridges this gap: it enables gradient-based optimization of delivery parameters and can be embedded as a differentiable layer within DL models for end-to-end training. While prior work explored PyTorch-based optimization for CyberKnife (Liang et al., 2022) and GPU-based proton dose computation (Bhattacharya et al., 2025), neither offered full VMAT physics with gradient access.

We present PyDoseRT, a modular, gradient-enabled dose engine in PyTorch, and demonstrate its use as a differentiable training component for DL-based treatment parameter prediction on the LUND-PROBE dataset.

## 2. Materials and Methods

### 2.1. Dose Engine

PyDoseRT is a physics-based pencil-beam dose engine that maps delivery parameters (MLC leaf/jaw positions, monitor units) to 3D dose distributions for a given patient CT and beam geometry. The pipeline consists of six layers: (1) fluence map generation from aperture parameters with physics corrections; (2) fluence projection into 3D with beam divergence and inverse-square law; (3) radiological depth computation via ray-tracing; (4) depth-dependent pencil-beam kernel construction (Nyholm et al., 2006); (5) grouped 2D convolution of fluence with kernels; (6) coordinate transformation and dose accumulation. Layers on optimizable parameters preserve gradients; fixed computations (radiological depth, kernels) run outside the gradient graph. Beam models were commissioned against water-tank measurements for an Elekta Versa HD and a Varian TrueBeam via an automated fitting pipeline.

### 2.2. Dose Engine Validation

The engine was validated on two prostate VMAT cohorts: 19 patients (Umeå, Gold Atlas (Nyholm et al., 2018), Varian TrueBeam, 42.7 Gy/7 fx) and 162 patients (Vienna, Elekta Versa HD, 60 Gy/20 fx). Clinical DICOM RTPLAN parameters were recalculated in PyDoseRT and compared against TPS doses using gamma analysis (2%/2 mm, 10% threshold, excluding 1 cm from external contour), isodose Dice coefficients, and mean absolute dose difference (MADD).

### 2.3. Deep Learning-Based Parameter Prediction

An attention-based DL model was trained on the LUND-PROBE dataset to predict machine parameters from patient CT, target, and organ-at-risk masks. Predicted MLC positions, jaw settings, and monitor units are passed through PyDoseRT during training; losses on the resulting dose backpropagate through the engine to the network weights, enabling end-to-end optimization without a commercial TPS.

## 3. Results

### 3.1. Dose Engine Validation

Water phantom depth-dose curves and lateral profiles showed close agreement with measurements (MAE $\leq 1.5\%$). Clinical recalculation results are summarized in Table 1. On an NVIDIA A40, a VMAT forward pass took $2.1\,\mathrm{s}$ and a forward-backward pass $5.1\,\mathrm{s}$.

Table 1: Dose engine validation: agreement between PyDoseRT and TPS (mean [range]).

| Metric | Umeå cohort | Vienna cohort | Unit |
|---|---|---|---|
| Gamma pass (2%/2 mm) | 99.64 [98.79–100.00] | 97.52 [89.64–99.76] | % |
| Dice (50% isodose) | 0.98 [0.98–0.99] | 0.98 [0.94–0.99] | – |
| Dice (95% isodose) | 0.96 [0.95–0.98] | 0.95 [0.84–0.98] | – |
| MADD | 0.07 [0.05–0.11] | 0.05 [0.02–0.11] | Gy |

### 3.2. Deep Learning-Based Parameter Prediction

Table 2 summarizes the DL model performance on the held-out LUND-PROBE validation set (83 patients). The model predicted deliverable machine parameters that, when recalculated with PyDoseRT, produced dose distributions in close agreement with clinical reference plans. Predictions can also be exported as DICOM RTPLANs.

Table 2: DL-predicted plan quality on the LUND-PROBE validation set (mean $\pm$ std).

| Metric | Value | Unit |
|---|---|---|
| Dice (50% isodose) | $0.87 \pm 0.02$ | – |
| Dice (95% isodose) | $0.92 \pm 0.03$ | – |
| MADD | $0.93 \pm 0.13$ | Gy |

## 4. Discussion and Conclusions

PyDoseRT demonstrates that physics-informed radiotherapy planning can be performed entirely within automatic differentiation frameworks. Unlike existing open-source engines (Wieser et al., 2017; Bhattacharya et al., 2025) that lack differentiability or GPU acceleration, and unlike RL-based approaches (Mekki et al., 2025; Achlatis et al., 2025) that require surrogate dose models, PyDoseRT enables direct gradient-based optimization of delivery parameters through physical dose computation. With forward-backward pass times of $\sim 5\,\mathrm{s}$ per VMAT plan, it supports iterative optimization within clinically relevant timeframes (Qiu et al., 2023). Current limitations include 2D pencil beam kernels, a focus on prostate treatments, and fixed gantry angles. The LUND-PROBE experiment demonstrates PyDoseRT's utility as a differentiable training layer for end-to-end DL-based planning. Code is available at https://github.com/UMU-DDI/PyDoseRT, with examples.

## Acknowledgments

We are grateful for the financial support obtained from the Cancer Research Foundation in Northern Sweden (AMP 24-1151). The computations were enabled by resources provided by the National Academic Infrastructure for Supercomputing in Sweden (NAISS), partially funded by the Swedish Research Council through grant agreement no. 2022-06725.

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
