# OpenReview forum: "PyDoseRT: A physics-informed, plug-and-play dose engine for gradient-based radiotherapy treatment planning"
_MIDL.io/2026/Short_Papers — MIDL 2026 - Short Papers Poster_

### Official Review · Reviewer_8VBa · 2026-05-03
**Interesting work with potential for impact**

**Rating:** 4
**Confidence:** 4

**Review:**

This is a very interesting paper with potential for impact in radiotherapy planning. While deep learning methods for dose prediction exist, translating them into machine-deliverable parameters often requires other approaches. By providing a differentiable pipeline, parameters can be optimized while optimizing dose delivery.

The paper is generally well-written and easy to follow. Code will be prepared by the authors.

W.r.t. performance, the authors mention that a forward and backward pass take 2.1 and 5.1 seconds, respectively. This seems slow, particularly when embedded in a deep learning approach. It would be good to give some indication of training and inference times.

**Summary:**

The paper describes a differentiable framework to compute 3D dose distributions from machine-deliverable parameters in radiotherapy. Authors evaluate the model by comparing it with existing approaches across a large number of cases and integrate the differentiable model into a deep learning pipeline that demonstrates the pipeline can help identify optimal radiotherapy settings.

**Strengths:**

- Very interesting application in radiotherapy.
- The paper addresses an unmet need in DL-based radiotherapy planning, where dose prediction has been addressed but parameter optimization has not.
- The paper will be of interest to the community.

**Weaknesses:**

- It would be good to get a bit more insight into the computational aspect of the work.
- It's unclear how many parameters exactly can be jointly optimized, and if there is a maximum for this.
- Results in Sec 3.3 look quite a bit worse than in Sec. 3.2. It would be good to gain some more insight into why this is the case.

**Justification Of Rating:**

Interesting work extending the capabilities of DL in radiotherapy towards true treatment optimization.

---

### Decision · Program_Chairs · 2026-05-08

Accept (Poster)